# Shift-Invariant Attribute Scoring for Kolmogorov-Arnold Networks via Shapley Value

## Abstract

For many real-world applications, understanding feature-outcome relationships is as crucial as achieving high predictive accuracy. While traditional neural networks excel at prediction, their black-box nature obscures underlying functional relationships. Kolmogorov–Arnold Networks (KANs) address this by employing learnable spline-based activation functions on edges, enabling recovery of symbolic representations while maintaining competitive performance. However, KAN's architecture presents unique challenges for network pruning. Conventional magnitude-based methods become unreliable due to sensitivity to input coordinate shifts. We propose **ShapKAN**, a pruning framework using Shapley value attribution to assess node importance in a shift-invariant manner. Unlike magnitude-based approaches, ShapKAN quantifies each node's actual contribution, ensuring consistent importance rankings regardless of input parameterization. Extensive experiments on synthetic and real-world datasets demonstrate that ShapKAN preserves true node importance while enabling effective network compression. Our approach improves KAN's interpretability advantages, facilitating deployment in resource-constrained environments.

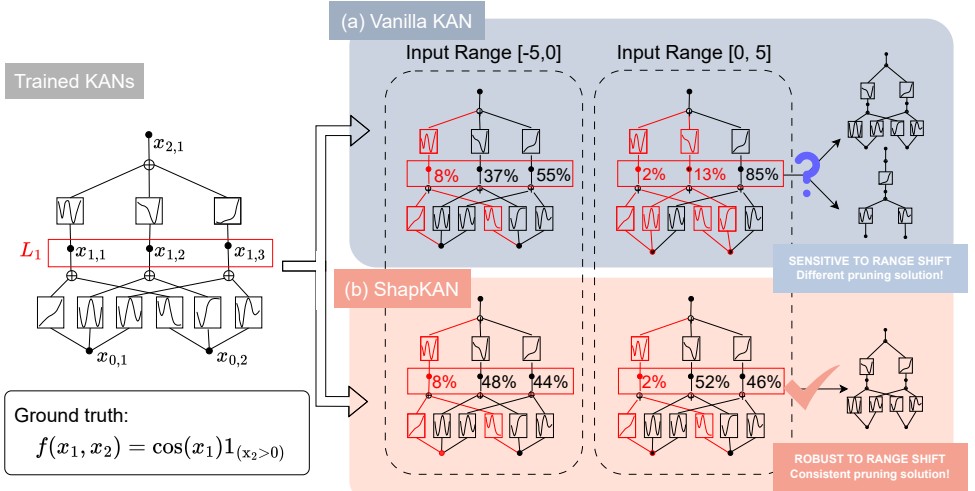

Figure 1: Example of neuron importance scoring of ShapKAN and Vanilla KAN under shifted domain. Scores are normalized to $[0, 1]$. (a): Vanilla KAN yields results with significant fluctuation. (b): ShapKAN provides more robust scoring consequence and consistent pruning strategy.

## 1 Introduction

Understanding feature-outcome relationships is often as crucial as achieving high predictive performance. In clinical research, for example, identifying which exposures drive treatment response

directly informs therapy decisions— providing actionable insights beyond survival prediction alone (Li et al., 2024). While traditional statistical models can reveal these relationships, they often rely on relatively strong parametric assumptions and are sensitive to model misspecification. On the other hand, neural networks offer much more flexibility, but suffer from a black-box nature that obscures the precise relationships between features and outcomes, limiting their utility for scientific inference.

Kolmogorov-Arnold Networks (KANs) (Liu et al., 2024b) fill in the research gap by leveraging learnable spline-based activation edge functions, enabling superior function approximation to ground truth with fewer parameters and better scaling laws behavior. Since the symbolic representation emerges from the composition of all learnable activation functions, a more compact KAN architecture facilitates a simpler and more interpretable symbolic function. Since KANs generally achieve better performance and interpretability at smaller scales, simplification techniques are crucial. Liu et al. (2024b) provide vanilla pruning method, which is applied by fallowing work.

However, existing pruning methods for KAN face key limitations. For example, PyKAN (Liu et al., 2024a) overwrites its cached data during each forward pass, causing the same function to yield inconsistent importance scores across domains, leading to unstable pruning results (see Section 4 and Figure 1). On the other hand, magnitude-based pruning methods evaluate neurons in isolation (Han et al., 2015), ignoring the compositional structure where univariate functions combine to approximate complex mappings. In addition, over-parameterized KANs are often trained to high accuracy before applying simplification, post-functional approaches such as pruning and symbolification remain underdeveloped due to KANs' unique architecture and recency. Although DropKAN (Altarabichi, 2025) is developed as an on-training regularization, principled post-training compression remains unresolved. Unlike traditional networks pruning scalar weights, KANs require attribution over continuous functions interacting with neurons, posing a fundamentally different challenge.

Shapley values (SV), originating from cooperative game theory (Shapley, 1953), quantify each player's marginal contribution across all coalitions. In KANs, pruning relies on reliable importance scores of neurons and edges: activations in one layer naturally form a cooperative game for the components in the next layer, making SV a natural criterion for node attribution. This motivates our investigation into Shapley value–guided pruning for KANs. However, the exponential growth of coalition subsets renders exact SV computation intractable for large networks. To address this challenge, efficient approximation methods are required to make SV estimation feasible and to extend it to multi-layer network architectures in practice (Rozemberczki et al., 2022; Maleki et al., 2013; Covert & Lee, 2021).

Our contributions are threefold: (1) We develop **ShapKAN**, an explainable framework based on Shapley values that quantifies neuron contributions in KAN layers and guides pruning; (2) We design efficient approximation strategies and a bottom-up multi-layer pruning algorithm, enabling scalable application with reduced computational cost; (3) Through extensive experiments on both simulated and real-world datasets, we show that ShapKAN yields consistent attribution scores under covariate shift while serving as a competitive model compression technique that preserves generalization capacity.

## 2 Preliminary

### 2.1 Kolmogorov-Arnold Networks (KANs)

Kolmogorov-Arnold Networks (KANs) are inspired by the Kolmogorov-Arnold representation theorem, which posits that any multivariate continuous function $f$ on a bounded domain can be decomposed into a finite sum of univariate functions (Kolmogorov, 1961; Braun & Griebel, 2009). The modern KAN architecture simplifies this by presenting a smooth function $f(\mathbf{x})$ as:

$$f(\mathbf{x}) = f(x_1, \ldots, x_n) = \sum_{i=1}^{2n+1} \Phi_i(\sum_{j=1}^{n} \phi_{i,j}(x_j)) \tag{1}$$

where $\phi_{i,j} : [0,1] \to \mathbb{R} \; \Phi_j : \mathbb{R} \to \mathbb{R}$. Intuitively, this motivates the use of B-spline basis functions in KANs to represent the trainable activation functions. Liu et al. (2024b) formalized KAN layers

where the output neurons $x_{l+1,j}$ is the sum of all related post-activations as :

$$x_{l+1,j} = \sum_{i=1}^{n_i} \phi_{l,j,i}(x_{l,i}), j = 1, ..., n_{l+1} \tag{2}$$

$\Phi$ is the matrix form of $\phi$ to different layer, where $x_{l+1,j}$ is the summation of post-activations. $x_{l+1,j}$ can be recognized as *cache data*, which is stored in the database of KANs model and automactically updated and saved during the forward process. Therefore, the output of a $L$ layers KAN with input vector $x_o \in R^{in}$ can be represented as

$$KAN(x) = (\Phi_{L-1} \circ \Phi_{L-2} \circ ... \circ \Phi_1 \circ \Phi_0)x \tag{3}$$

Through pruning, a KAN can be simplified into a symbolic function, which can act as as interpretable ground truth.

## 2.2 CHALLENGES IN PRUNING AND COMPRESSION FOR KANS

KANs face unique challenges in pruning and model compression. Traditional methods - such as magnitude-based sparsification (Han et al., 2015), sparse subnetwork identification (Frankle & Carbin, 2019; Lee et al., 2018), and greedy or initialization-driven strategies (Resende & Ribeiro, 2016; Wang et al., 2020; Tanaka et al., 2020; Hoefler et al., 2021) - are conceptually designed for scalar weights or neurons. They are therefore mismatched for KANs, where the fundamental unit is the activation function on edges. The challenge of KANs pruning is not simply to find a sparse set of weights, but to assess the contribution of an entire function. In the original KANs pruning method, sparsity is performed through L1 regularization and entropy regularization, where

$$\mathcal{L} = \mathcal{L}_{\text{pred}} + \lambda \left( \mu_1 \sum_{l=0}^{L-1} \|\Phi_l\|_1 + \mu_2 \sum_{l=0}^{L-1} \|S(\Phi_l)\| \right), \tag{4}$$

$\mathcal{L}_{pred}$ is the data loss, $\lambda$ is the overall penalty, $\mu_1, \mu_2$ are weighting factors (typically $\mu_1 = \mu_2 = 1$), and $S(\Phi_l)$ is a sparsity-inducing transformation of $\Phi_l$, which is the summation of activation functions. This penalty train irrelevant nodes toward sparsification. The incoming and outgoing pruning score are defined as:

$$I_{l,i} = \max_k(|\phi_{l-1,i,k}|_1), O_{l,i} = \max_j(|\phi_{l-1,i,k}|_1) \tag{5}$$

and nodes are retained only if $I_{l,i}$ and $O_{l,i}$ exceed a threshold hyperparameter ($\theta = 10^{-2}$ in default). This procedue encourages sparsity and enables pruning. (Liu et al., 2024b) Beyond this baseline, Altarabichi (2025) proposes *DropKAN*, the only existing work related to pruning KANs. It uses masking-based pruning strategies to address gradient vanish while some neuron is masking. While DropKAN improves efficiency, it directly transplant the Dropout method from magnitude-based models, which lead to pruning scores unprincipled. Furthermore, it did not perform well in our real-world experimental evaluations. Other KAN-related packages, including *efficient-kan* (Cacciatore et al., 2024), *fast-kan* (Li, 2024), and *torch-kan* (Bhattacharjee, 2024), do not address pruning directly. Therefore, leave a gap here.

## 2.3 SHAPLEY VALUE

The Shapley value (SV) (Shapley, 1953) originates from cooperative game theory, where a grand coalition $D$ of players jointly generates a total profit $v(D)$. The SV provides a principled way to allocate this profit fairly among players by quantifying the average marginal contribution of each player $i \in D$ across all subsets $S \subseteq D \setminus \{i\}$:

$$\text{Shap}(i) = \frac{1}{|D|} \sum_{S \subseteq D \setminus \{i\}} \binom{|D| - 1}{|S|}^{-1} (v(S \cup \{i\}) - v(S)), \tag{6}$$

In machine learning, two value functions are common: the *prediction game*, where $v(S) = \mathbb{E}_{\mathbf{x}}[f_S(\mathbf{x})]$, and the *validation (loss) game*, $v(S) = -\mathbb{E}_{\mathbf{x}}[\ell(f_S(\mathbf{x}), \mathbf{y})]$, where $\mathbf{y}$ denotes the true

labels. While the validation game exploits more information by incorporating labels, it is inapplicable for unlabeled data, making the prediction game often the only option. (Rozemberczki et al., 2022; Lundberg & Lee, 2017; Covert et al., 2020; Ghorbani & Zou, 2020).

As shown in equation 6, computing SV exactly requires evaluating exponentially many subsets, which is computationally infeasible for large $D$. Approximation methods include: (1) *Permutation sampling* (Castro et al., 2009; Mitchell et al., 2022); (2) *Weighted least-squares optimization* (the kernel method) (Lundberg & Lee, 2017; Covert & Lee, 2021); (3) *Amortized explanation method* (Jethani et al., 2021; 2022). Among these, permutation sampling benefits from the Monte Carlo estimator and is provably unbiased, converging asymptotically at the rate $O(1/\sqrt{n})$ (Castro et al., 2009; Maleki et al., 2013). Compared to permutation sampling, the kernel method and amortized approaches generally rely on narrower theoretical assumptions and often introduce additional requirements or auxiliary components (e.g., a linear constraint), which reduces flexibility. In addition, amortized methods typically involve a black-box explainer (i.e., a multilayer perceptron), further limiting their applicability.

## 3 METHODOLOGY

### 3.1 SHAPKAN: FORMALIZATION AS A COOPERATIVE GAME

In Section 2, we discussed that the output neurons of KANs are formed by summing all relevant post-activations. Hence, the final model output can be represented as a nested summation over neurons, where each neuron is activated by functions defined on edges. This additive property naturally corresponds to a cooperative game, where neurons collaborate layer by layer to contribute to the prediction power of a KAN model on a fixed dataset. Formally, we denote the index set of neurons (nodes) in layer $l$ as

$$N_l = \{1, 2, \ldots, n_l\}, \quad L_l = \{x_{l,i} : i \in N_l\}.$$

For any subset of indices $S_l \subseteq N_l$, we denote the corresponding subset of neurons as $S_{L_l} = \{x_{l,i} : i \in S_l\}$. In the cooperative game defined at layer $l$, the value function for a coalition $S_{L_l}$ under the prediction game is given by

$$v(S_{L_l}) = \mathbb{E}_{\mathbf{x}} \left[ KAN_{S_{L_l}}(\mathbf{x}) \right], \tag{7}$$

where $KAN_{S_{L_l}}(\mathbf{x})$ indicates that for each $j = 1, \ldots, n_{l+1}$,

$$x_{l+1,j} = \sum_{i \in S_l} \phi_{l,j,i}(x_{l,i})$$

within the KAN computation graph.

Overall, ShapKAN models neuron attribution as a cooperative game with the following components:

- **Players:** All neurons in $L_l$ in layer $l$ of the KAN model.
- **Coalitions:** Any subset of neurons $S_{L_l} \subseteq L_l$.
- **Value function:** $v(S_{L_l})$ defined in equation 7.

As originally proposed by Shapley (1953), the Shapley value satisfies several fairness axioms, which have since been widely adopted in machine learning explanation methods (Rozemberczki et al., 2022; Lundberg & Lee, 2017; Ghorbani & Zou, 2020). ShapKAN inherits these desirable properties:

- **Dummy (Null player):** If a neuron $x_{l,i}$ contributes nothing to any coalition, i.e.,

$$\forall S_{L_l} \subseteq L_l \setminus \{x_{l,i}\} : \quad v(S_{L_l} \cup \{x_{l,i}\}) = v(S_{L_l}),$$

  then $\text{Shap}(x_{l,i}) = 0$. For instance, this holds when a neuron has all-zero parameters.
- **Efficiency:** The attributions sum to the total value:

$$\sum_{i \in N_l} \text{Shap}(x_{l,i}) = v(N_l) - v(\emptyset).$$

  Since removing all nodes yields an empty model, we have $v(\emptyset) = 0$, and thus the sum of Shapley values equals $v(N_l)$ according to equation 6.

- **Symmetry:** If two neurons $x_{l,i}$ and $x_{l,j}$ contribute equally to all coalitions, then

$$\mathrm{Shap}(x_{l,i}) = \mathrm{Shap}(x_{l,j}).$$

Formally, if

$$\forall S_{L_l} \subseteq L_l \setminus \{x_{l,i}, x_{l,j}\} : \quad v(S_{L_l} \cup \{x_{l,i}\}) = v(S_{L_l} \cup \{x_{l,j}\}),$$

then their Shapley values are identical.

Capitalizing on the inherent fairness axioms above, ShapKAN is able to fairly evaluate and quantify the importance of neurons, and is especially robust in situations where covariates are shifting. As demonstrated in Section 4, the attribution score is shift-invariant in the ShapKAN framework, which will benefit the effectiveness and interpretability of KANs in various scientific domains.

## 3.2 APPROXIMATION METHOD FOR SHAPKAN

As discussed in Section 2.3, computation of Shapley values requires enumerating all $2^{|N_l|}$ coalitions, which is infeasible for practical models. To address this challenge, approximation strategies have been developed. In ShapKAN, we adopt *permutation sampling* together with its variance reduction technique, owing to their unbiasedness and generality.

Permutation sampling estimates the Shapley value of neuron $x_{l,i} \in L_l$ by averaging its marginal contributions across a batch of randomly sampled permutations:

$$\widehat{\mathrm{Shap}}(x_{l,i}) = \frac{1}{m} \sum_{t=1}^{m} \left( v(S_{L_l}^{(t)} \cup \{x_{l,i}\}) - v(S_{L_l}^{(t)}) \right), \tag{8}$$

where $\{S_{L_l}^{(t)}\}_{t=1}^{m}$ are subsets induced by a uniformly sampled set $\Pi \subseteq \sigma_{N_l}$ of $m$ permutations of indices $N_l = \{1, \ldots, n_l\}$. For each permutation $\sigma^{(t)} \in \Pi$, $S_{L_l}^{(t)}$ denotes the set of neurons that precede $x_{l,i}$. This Monte Carlo estimator is unbiased and converges asymptotically at the standard rate $O(1/\sqrt{m})$ by the central limit theorem (Castro et al., 2009). In implementation, each permutation $\sigma$ is encoded as a binary mask $Z \in \{0, 1\}^{n_l}$, where $Z_i = 1$ indicates that $x_{l,i}$ is included in the current coalition $S_{L_l}$.

To further reduce variance, we incorporate *antithetic permutation sampling* (Mitchell et al., 2022), which pairs each sampled permutation $\sigma^{(t)}$ with its antithetic counterpart $A(\sigma^{(t)})$ (e.g. the reverse of $\sigma^{(t)}$). The corresponding estimator is

$$\widehat{\mathrm{Shap}}(x_{l,i}) = \frac{1}{m} \sum_{t=1}^{m} \frac{\left( v(S_{L_l}^{(t)} \cup x_{l,i}) - v(S_{L_l}^{(t)}) \right) + \left( v(S_{L_l}^{(t),\mathrm{anti}} \cup x_{l,i}) - v(S_{L_l}^{(t),\mathrm{anti}}) \right)}{2}, \tag{9}$$

where for each permutation $\sigma^{(t)}$, $S_{L_l}^{(t)}$ is the set of neurons preceding $x_{l,i}$, and $S_{L_l}^{(t),\mathrm{anti}}$ is the set induced by its antithetic permutation $A(\sigma^{(t)})$. As shown in Lomeli et al. (2019), this estimator is also unbiased and achieves lower variance, particularly in the small-sample regime, thereby reducing the number of required permutations and accelerating estimation.

By leveraging these sampling schemes, ShapKAN achieves efficient and statistically reliable estimation of neuron importance without relying on restrictive model assumptions. Detailed results are provided in Section 4.

## 3.3 PRUNING STRATEGY FOR MULTIPLE LAYERS

In Sections 3.1 and 3.2, we formulated Shapley value computation for neurons in a single KAN layer and proposed efficient approximation techniques. Since each neuron's contribution propagates through subsequent layers via the compositional structure of KANs (see equation 2), pruning decisions at one layer influence the entire network.

KANs are typically constructed with multiple layers (Liu et al., 2024b;a), where the number of parameters decreases progressively from bottom to top, consistent with the engineering heuristic that higher layers capture more abstract representations. Motivated by this structure, we adopt a

*bottom-up greedy pruning algorithm*: Shapley values are estimated layer by layer, and neurons with low importance are pruned sequentially from the bottom layer upwards.

For SV calculation, we distinguish between small and wide layers. When the width of a layer is small (e.g., $n_l < 8$), the exact Shapley values can be computed exhaustively (see Section 3.1). For wider layers, we approximate SVs until either a convergence criterion (tiny changes detected) is satisfied or a sufficiently large number of permutations has been sampled (see Section 3.2).

To support different application scenarios, we design flexible pruning criteria: (1) *Ratio pruning:* Remove neurons whose absolute SV falls below a given ratio relative to the total contribution in that layer; (2) *Number pruning:* Remove a user-specified number of neurons with the smallest SV in each layer; (3) *Threshold pruning:* Remove neurons whose SV is below a specified threshold; thresholds can differ across layers.

The complete ShapKAN pruning procedure for multi-layer KANs is summarized in Algorithm 1. Our experiments demonstrate the effectiveness of this framework and its improvement on generalization compared to vanilla pruning method in Section 5. We avoid applying SV scoring directly to the layer corresponding to input features. This is because, in the literature on SV–based feature importance attribution, removing features from coalitions requires modeling conditional distributions of inputs. Consequently, applying ShapKAN at the first layer would essentially reduce to existing model-agnostic SV methods for feature attribution (Rozemberczki et al., 2022; Lundberg & Lee, 2017).

---

**Algorithm 1** ShapKAN Multi-Layer Pruning Framework

---

**Require:** KAN model $M$, dataset $D$, pruning criterion (ratio $\eta$, number $k$, or threshold $\tau$)
**Ensure:** Pruned KAN model $M'$
1: $M' \leftarrow M.\text{copy}()$             $\triangleright$ Initialize working model
2: **for** $l = 1$ **to** $L - 1$ **do**
3:   $N_l \leftarrow \{1, 2, \ldots, n_l\}$         $\triangleright$ Index set of neurons in layer $l$
4:   **if** $|N_l| < 8$ **then**
5:    $Shap^{(l)} \leftarrow \text{EXACTSHAPLEY}(M', D, l)$     $\triangleright$ Small layer: exact SV
6:   **else**
7:    $Shap^{(l)} \leftarrow \text{APPROXSHAPLEY}(M', D, l)$    $\triangleright$ Wide layer: approximation
8:   **end if**
9:   $S_{\text{prune}} \leftarrow \text{SELECTNODES}(Shap^{(l)}, \eta, k, \tau)$    $\triangleright$ Identify low-SV neurons
10:   $M' \leftarrow \text{PRUNE}(M', l, S_{\text{prune}})$      $\triangleright$ Remove selected neurons
11: **end for**
12: **return** $M'$

---

## 4 SIMULATION STUDIES

Following the experimental protocol in Liu et al. (2024a), we evaluate proposed method against baseline under covariate shift focusing on model performance for both *prediction tasks* (measuring model accuracy) and *non-prediction tasks* (measuring interpretability through recovery of ground-truth symbolic functions). We compare the effectiveness of ShapKAN against the vanilla KAN pruning method across these task types in shifting scenarios[1].

### 4.1 SIMULATION SETUP

We adopt four synthetic datasets proposed in the original KAN paper (Liu et al., 2024a), each designed to capture distinct mathematical or physical characteristics:

- **Multiplication (bilinear interactions):** $f_1(x_1, x_2) = x_1 x_2$
- **Special Function (high-frequency oscillations with Bessel functions)[2]:** $f_2(x_1, x_2) = \exp\big(J_0(20x_1) + x_2^2\big)$

---

[1]Our code is available in an anonymized repository `https://anonymous.4open.science/r/ShapKAN`, which will be published after review.

[2]$J_0$ denotes the Bessel function of the first kind of order 0.

- **Phase Transition (sharp transitions near constraint manifolds):** $f_3(x_1, x_2, x_3) = \tanh\left(5\left(\sum_{i=1}^{3} x_i^4 - 1\right)\right)$

- **Complex Function (multi-scale periodic and exponential behavior):** $f_4(x_1, x_2) = \exp\left(\sin(\pi x_1) + x_2^2\right)$

The KAN architecture is fixed as $[d, 5, 1]$, where $d$ denotes the input dimension, 5 is the hidden layer width and 1 is the output dimension, with B-spline activation functions of degree 3. Training data are sampled from $\mathcal{N}(0, 1)$ with random seeds, restricted to input ranges $[-1, 1]$. Test data are generated in three ranges: $[-1, 1]$, $[-1, 0]$, and $[0, 1]$, to simulate covariate shift. Each experiment is repeated for 50 independent runs to evaluate the stability of attribution scores and their effect on pruning. Model hyperparameters and training details are provided in the Appendix A.2.

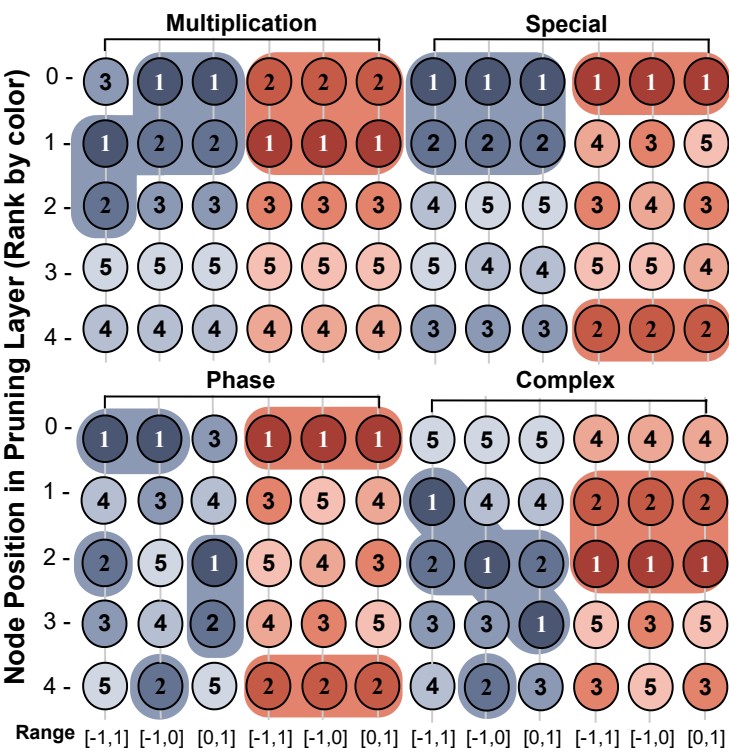

Figure 2: Cross-domain robustness comparison of neuron importance ranking between Vanilla KAN and ShapKAN. Each circle represents a node's importance rank evaluated by scoring methods. Shadow indicates the most substantial nodes. Numeric record of mean, standard deviations and pruning set are reported in the Appendix A.1.1.

## 4.2 NEURON IMPORTANCE SCORES

We first compare neuron importance scores estimated by ShapKAN and the baseline Vanilla KAN. For fair comparison, scores are normalized into percentages. Figure 2 reports results on both balanced and shifted test data. ShapKAN maintains consistent ranking distributions, whereas Vanilla KAN exhibits large variations. Moreover, ShapKAN shows smaller standard deviations and range differences in scores, suggesting more stable estimation (see Appendix A.1.1).

To assess pruning effectiveness, we prune the same number of least-important neurons under both methods to maintain identical parameter counts. For example, in the Special task, Vanilla KAN keeps neurons $[0, 1]$ and prunes $[2, 3, 4]$, while ShapKAN instead prunes $[1, 2, 3]$. After pruning, models are retrained with identical settings, and we report test RMSE corresponding to the best

training epoch, following Han et al. (2015). The details of pruning decision of other datasets are provided in Appendix A.1.1. ShapKAN consistently achieves lower test RMSE than Vanilla KAN, demonstrating that Shapley-based attribution enables pruning decisions that better preserve predictive performance and generalization.

### 4.3 SYMBOLIC REGRESSION FOR RECOVERING THE GROUND-TRUTH

We further test interpretability by reproducing the symbolic regression setup from Liu et al. (2024b), extending it to scenarios with covariate shift. Figure 3 shows results on the multiplication dataset with data for neuron scoring restricted to $[0, 1]$. ShapKAN more accurately recovers the ground-truth formulation, whereas Vanilla KAN yields spurious expressions. With prior knowledge of function structure, the pruning strategy slightly differs from Section 4.2, yet ShapKAN remains more robust. Additional symbolic regression results are provided in the Appendix A.1.2.

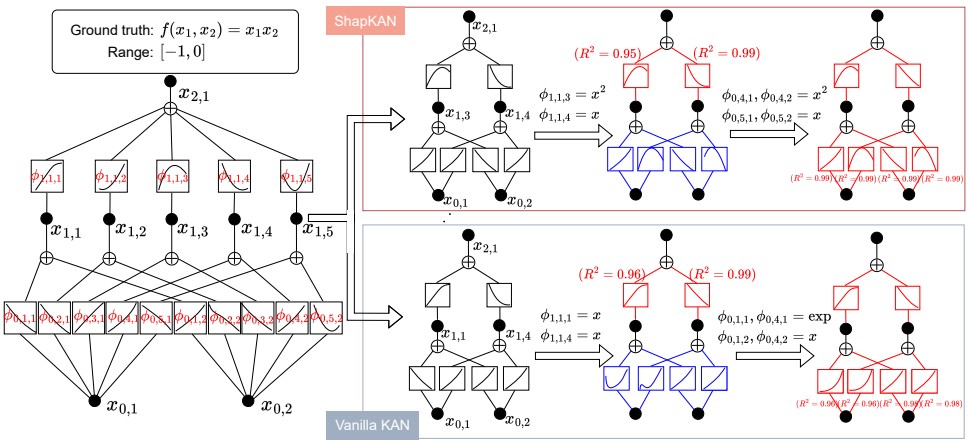

Figure 3: Symbolic regression comparison on the multiplication dataset under covariate shift. $R^2$ measures symbolic similarity (1 = exact match). ShapKAN recovers $\hat{f}(x_1, x_2) \approx x_1 x_2 + c$ (small constant), while Vanilla KAN yields $\hat{f}(x_1, x_2) \approx -0.01x_2 + 0.01e^{x_1} + c$. Red edges denote validated symbolic functions, blue edges represent refitted functions.

### 4.4 PERMUTATION SAMPLING

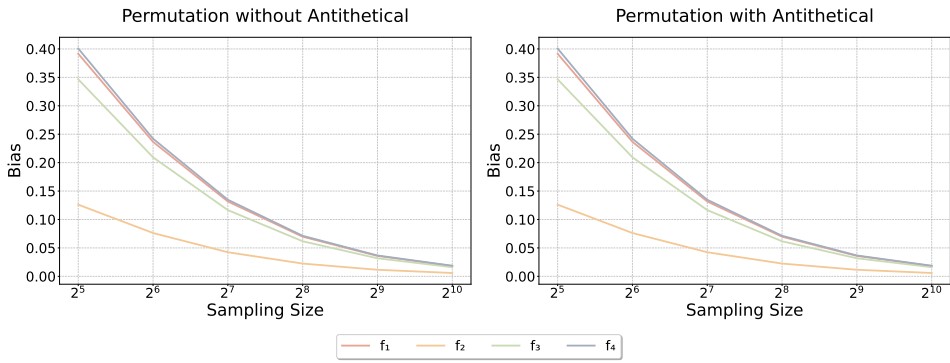

Figure 4: Convergence analysis of permutation sampling methods on simulated datasets. Left: Standard permutation sampling. Right: Antithetical permutation sampling.

As discussed in Section 3.2, we implement permutation sampling with optional antithetic pairs for wide KAN layers, where the calculation of actual SV is intractable. To validate correctness, we record the bias (measured as $\ell_2$ distance between estimated and exact SVs) under varying sampling sizes. As shown in Figure 4, the bias decreases rapidly as the number of permutations grows.

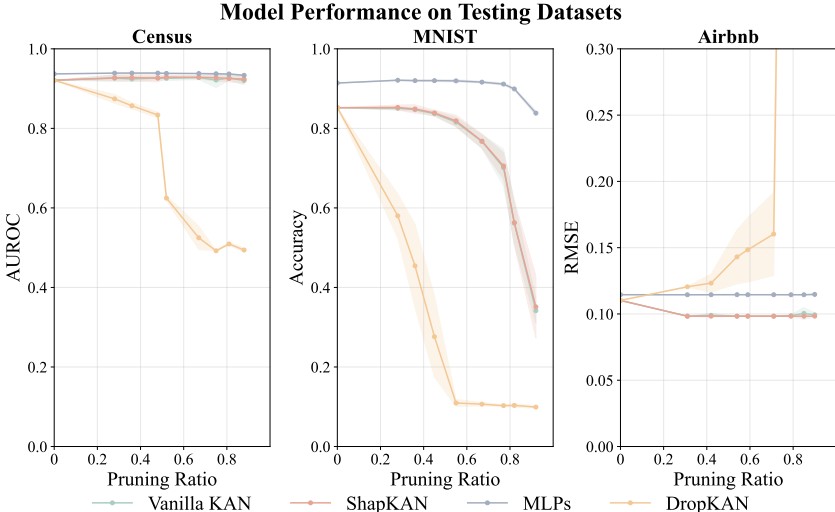

Figure 5: Comparison of generalization capacity. Mean and standard deviation are reported based on 10 times independent experiments. For accuracy and Area under the Receiver Operating Characteristic Curve (AUROC), higher are better; for RMSE, lower is better. In Airbnb dataset, DropKAN's RMSE significantly exceeds 0.3 when the ratio is above 0.6.

In practice, ranking stability matters more than absolute bias. We observe that ranking distributions converge quickly even with moderate sampling sizes, ensuring reliable pruning guidance. Moreover, runtime analysis with larger sampling sizes, showing minimal overhead (see Appendix A.1.3).

## 5  REAL-WORLD BENCHMARKS

To further evaluate ShapKAN's effectiveness across diverse domains, we conduct experiments on three representative ML datasets: **Census-income**(tabular classification), **MINST** (computer vision), and **Airbnb** (mixed-type regression) over different score pruning rates. We compare Shap-KAN against the vanilla KAN, DropKAN, and MLP baselines under varying pruning ratios. For fairness, the MLP baseline is configured with a similar number of parameters as KAN. Implementation details and hyperparameters are provided in Appendix A.2.

As illustrated in Figure 5, ShapKAN consistently outperforms Vanilla KAN in generalization. Drop-KAN shows the poorest performance across all three datasets, confirming that informed pruning is essential. While the MLP baseline demonstrates strong performance on every task (consistent with Yu et al. (2024)), ShapKAN achieves competitive accuracy while additionally providing a crucial advantage: the ability to recover ground truth symbolic functions, as demonstrated in Section 4.

## 6  DISCUSSION

Building on Shapley values, ShapKAN provides an interpretable neuron scoring and pruning framework with a desirable shift-invariant property, outperforming the default magnitude-based methods in KANs. By offering stable and principled importance estimates, ShapKAN enhances confidence in the outcomes of KAN models for both prediction and non-prediction tasks, potentially facilitating their broader adoption in domains such as AI+Science, AI+Health, and AI+Finance.

In KAN 2.0 (Liu et al., 2024a), subnodes are introduced between node layers and edges to capture more complex symbolic relationships, while still preserving the additive property in equation 2. This makes ShapKAN readily applicable to newer KAN variants. For future work, we plan to investigate more heuristic algorithms and extend ShapKAN to account for higher-order Shapley interactions (Muschalik et al., 2024). Such extensions would allow more nuanced quantification of neuron contributions in KANs and further strengthen their interpretability.

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

# A APPENDIX

## A.1 DETAILS OF SIMULATION STUDIES

### A.1.1 NEURON IMPORTANCE SCORES SUPPLEMENT

In the main paper we visualized neuron importance rankings, but here we additionally report the numerical mean and standard deviation of the experimental results. Table 1 highlights the two most important neurons in red and blue, respectively. While the ranking of the remaining three neurons may change, their ShapKAN scores are consistently small, with mean values below 15%. By contrast, Vanilla KAN exhibits much larger fluctuations. For example, in the Special Function task, the score of the last neuron varies substantially from 13% to 27% and 32% across different ranges.

Table 1: Cross-domain robustness of neuron importance in Vanilla KAN and ShapKAN. Scores are normalized to percentages ($[0, 1]$). The most important neuron in each task is marked in red and the secondary one in blue. SVs have been normalized to percentages across 50 independent runs, and we report the mean values, resulting in summation that does not equal to exactly 100%.

| Function | Range | Vanilla KAN Node Importance ($\pm$ SD) | ShapKAN Node Importance ($\pm$ SD) |
|---|---|---|---|
| **Multiplication** ($f_1$) | $[-1, 1]$ | [0.19±0.031, 0.45±0.033, 0.23±0.038, 0.04±0.014, 0.09±0.021] | [0.27±0.012, 0.49±0.009, 0.14±0.008, 0.01±0.007, 0.09±0.006] |
| | $[-1, 0]$ | [0.56±0.002, 0.28±0.007, 0.27±0.002, 0.06±0.001, 0.15±0.002] | [0.21±0.001, 0.55±0.002, 0.14±0.000, 0.02±0.000, 0.09±0.000] |
| | $[0, 1]$ | [0.48±0.002, 0.34±0.009, 0.33±0.002, 0.06±0.001, 0.14±0.001] | [0.23±0.001, 0.55±0.002, 0.13±0.001, 0.01±0.000, 0.08±0.001] |
| **Special Function** ($f_2$) | $[-1, 1]$ | [0.38±0.079, 0.29±0.105, 0.13±0.107, 0.07±0.036, 0.13±0.112] | [0.53±0.094, 0.07±0.051, 0.09±0.052, 0.02±0.012, 0.29±0.062] |
| | $[-1, 0]$ | [0.60±0.013, 0.57±0.007, 0.01±0.000, 0.15±0.006, 0.27±0.005] | [0.46±0.01, 0.14±0.005, 0.07±0.0007, 0.01±0.001, 0.32±0.003] |
| | $[0, 1]$ | [0.74±0.012, 0.47±0.005, 0.03±0.001, 0.05±0.001, 0.32±0.003] | [0.59±0.004, 0.02±0.005, 0.06±0.001, 0.02±0.001, 0.31±0.002] |
| **Phase transition** ($f_3$) | $[-1, 1]$ | [0.29±0.015, 0.19±0.007, 0.21±0.007, 0.21±0.009, 0.09±0.007] | [0.62±0.011, 0.07±0.008, 0.01±0.007, 0.07±0.006, 0.23±0.005] |
| | $[-1, 0]$ | [0.53±0.008, 0.25±0.007, 0.15±0.005, 0.16±0.007, 0.26±0.003] | [0.45±0.003, 0.12±0.003, 0.12±0.003, 0.14±0.005, 0.18±0.004] |
| | $[0, 1]$ | [0.40±0.011, 0.31±0.010, 0.47±0.008, 0.46±0.006, 0.09±0.002] | [0.59±0.011, 0.06±0.005, 0.09±0.007, 0.04±0.004, 0.22±0.003] |
| **Complex function** ($f_4$) | $[-1, 1]$ | [0.02±0.004, 0.28±0.007, 0.25±0.012, 0.25±0.029, 0.20±0.041] | [0.07±0.009, 0.32±0.031, 0.45±0.011, 0.01±0.009, 0.15±0.039] |
| | $[-1, 0]$ | [0.10±0.003, 0.32±0.005, 0.51±0.014, 0.36±0.005, 0.46±0.003] | [0.13±0.001, 0.20±0.003, 0.53±0.005, 0.13±0.004, 0.03±0.005] |
| | $[0, 1]$ | [0.04±0.001, 0.26±0.007, 0.38±0.007, 0.45±0.005, 0.26±0.002] | [0.07±0.000, 0.37±0.004, 0.38±0.004, 0.06±0.005, 0.13±0.002] |

Based on the scores in Table 1, we prune neurons while keeping the parameter counts identical between Vanilla KAN and ShapKAN. For ShapKAN, the top neurons remain stable across covariate shifts, making pruning straightforward and interpretable. For Vanilla KAN, the scores differ substantially across ranges, so we retain neurons consistently ranked as important.

As shown in Table 2, we report the mean and standard deviation of test RMSE corresponding to the lowest training loss after pruning. ShapKAN consistently achieves lower mean RMSE and smaller variance, demonstrating that Shapley-based attribution leads to pruning decisions that better preserve predictive performance and generalization capacity.

Table 2: Comparison of pruning decisions guided by neuron importance rankings in Vanilla KAN and ShapKAN across tasks. Reported values are test RMSE $\pm$ standard deviation (lower is better).

| Task | Method | Pruned Neurons | Test RMSE $\pm$ Std Dev |
|---|---|---|---|
| Multiplication ($f_1$) | Vanilla | [0, 3, 4] | 0.00055 ± 0.00008 |
| | ShapKAN | [2, 3, 4] | **0.00031 ± 0.00005** |
| Special ($f_2$) | Vanilla | [2, 3, 4] | 0.558 ± 0.030 |
| | ShapKAN | [1, 2, 3] | **0.536 ± 0.013** |
| Phase ($f_3$) | Vanilla | [1, 3, 4] | 0.129 ± 0.041 |
| | ShapKAN | [1, 2, 3] | **0.109 ± 0.013** |
| Complex ($f_4$) | Vanilla | [0, 2, 3, 4] | 0.619 ± 0.015 |
| | ShapKAN | [0, 1, 3, 4] | **0.612 ± 0.012** |

### A.1.2 SYMBOLIC REPRESENTATION SUPPLEMENT

Following the symbolic regression experiments in KANs with human interaction (Liu et al., 2024b), we prune trained KAN models using neuron scores evaluated under covariate shift. Since the ground-truth functional structures of the simulated datasets are already known, we fix the number of remaining nodes as in Liu et al. (2024b). Notably, symbolic regression requires human interaction with one fixed model, making it infeasible to conduct repeated randomized experiments as in Appendix A.1.1.

Tables 3 and 4 present the symbolic regression process under covariate shift. In the Multiplication and Phase tasks, ShapKAN more faithfully recovers the ground-truth formulations. In the Special and Complex tasks, both ShapKAN and Vanilla KAN identify the same most important neurons, thereby producing the same correct symbolic expressions.

| Task | Range | Mode | Prune Set | $\Phi_1 (R^2)$ | $\Phi_2 (R^2)$ |
|---|---|---|---|---|---|
| Multiplication ($f_1$) | [-1,0] | ShapKAN | [0,1,4] | $x^2(0.95),\ x(0.99)$ | $x(0.99),\ x^2(0.99),\ x(0.99),\ x^2(0.99)$ |
| | | Vanilla KAN | [1,2,4] | $x(0.96),\ x(0.99)$ | $\exp(0.96),\ \exp(0.96),\ x(0.98),\ x(0.98)$ |
| Special ($f_2$) | [-1,0] | ShapKAN | [1,2,3,4] | $J_0(0.72),\ x^2(0.93)$ | $\exp(0.99)$ |
| | | Vanilla KAN | | | |
| Phase ($f_3$) | [0,1] | ShapKAN | [1,2,3,4] | $x^4(0.99),\ x^4(0.94),\ x^4(0.97)$ | $\tanh(0.99)$ |
| | | Vanilla KAN | [0,1,3,4] | $x^4(0.99),\ x^4(0.99),\ \sin(0.48)$ | $x^2(0.99)$ |
| Complex ($f_4$) | [0,1] | ShapKAN | [0,1,2,3] | $\sin(0.99),\ x^2(0.99)$ | $\exp(0.99)$ |
| | | Vanilla KAN | | | |

Table 3: Symbolic function identification result across KAN layers after pruning. Models trained on [-1,1] and tested on shift-domain. $R^2$ measure symbolic fitness for each activation functions in $(\Phi_1, \Phi_2)$.

| Task | Range | Mode | Prune Set | Symbolic Function |
|---|---|---|---|---|
| Multiplication($f_1$) | [-1,0] | ShapKAN | [0,1,4] | $\hat{f}_1(x_1, x_2) \approx x_1 x_2 + c$ |
| | | Vanilla KAN | [1,2,4] | $\hat{f}_1(x_1, x_2) \approx -0.01 x_2 + 0.01 e^{x_1} + c$ |
| Special($f_2$) | [-1,0] | ShapKAN | [1,2,3,4] | $\hat{f}_2(x_1) \approx exp(J_0(9.91x_1) + x_2^2 + c$ |
| | | Vanilla KAN | | |
| Phase($f_3$) | [0,1] | ShapKAN | [1,2,3,4] | $\hat{f}_3(x_1, x_2, x_3) \approx 0.8 \tanh(5x_3^4 + 1.6x_1^4 + 2x_2^4 - 4) + c$ |
| | | Vanilla KAN | [0,1,3,4] | $\hat{f}_3(x_1, x_2, x_3) \approx 0.4(0.3(0.3 - x_2)^4 + x_1^4 - 0.2\sin(7x_3 + 5) + 1)^2 + c$ |
| Complex($f_4$) | [0,1] | ShapKAN | [0,1,2,3] | $\hat{f}_4(x_1, x_2) = e^{x_2^2 - \sin(3.1x_1) + 3.14} + c$ |
| | | Vanilla KAN | | |

Table 4: Symbolic function recovery comparison between ShapKAN and Vanilla KAN.

### A.1.3 PERMUTATION SAMPLING SUPPLEMENT

Table 5 and 6 report the runtime overhead on the four simulated datasets. The results show that permutation sampling with the antithetic technique incurs negligible extra cost, even for large sampling sizes, compared to the time required for exact Shapley value computation. This demonstrates its practicality for real applications.

Beyond bias, practitioners often care more about the ranking distribution of estimated SVs. As illustrated in Figure 6, the approximation converges toward the ground-truth SV ranking as the sampling size increases, confirming that permutation sampling provides reliable estimates.

**Validation of Shapley Value Approximation Accuracy Across Different Sampling Sizes**

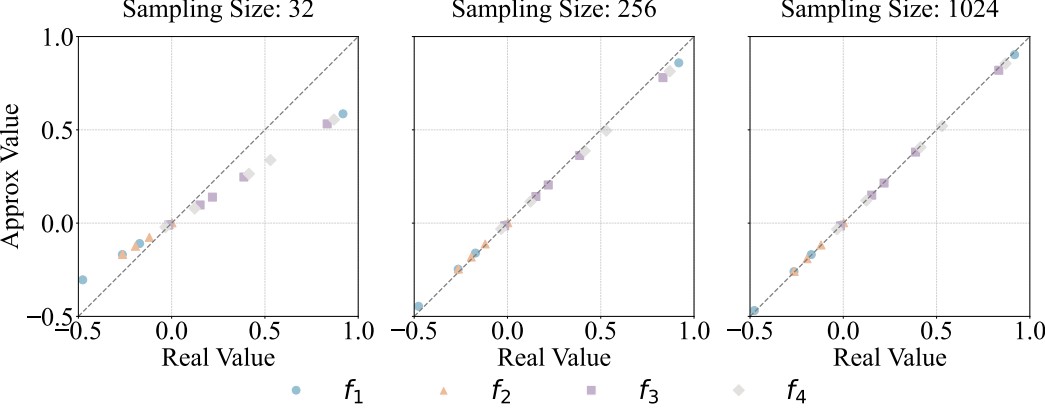

Figure 6: Validation of SV approximation accuracy against ground-truth values across different sampling sizes. As bias decreases, points align more closely with the diagonal line.

| Task | Computation Time (seconds) |
|---|---|
| Multiplication ($f_1$) | $0.69 \pm 0.14$ |
| Special Function ($f_2$) | $0.62 \pm 0.11$ |
| Phase Transition ($f_3$) | $0.62 \pm 0.08$ |
| Complex Function ($f_4$) | $0.60 \pm 0.17$ |

Table 5: Computational efficiency of exact Shapley values calculation across simulated datasets. Result show mean execution time and standard deviation over 20 independent runs.

Table 6: Computational efficiency of approximation methods. Reported values are average computation times in seconds ($\pm$ standard deviation) for SV estimation across four synthetic functions under varying sample sizes. Results are averaged over 20 independent runs. Standard permutation sampling and its antithetic variant are compared across increasing sampling sizes (32–1024) to assess scalability.

| Task | Permutation sampling size | | | | | |
|---|---|---|---|---|---|---|
| | 32 | 64 | 128 | 256 | 512 | 1024 |
| Multiplication | $0.60 \pm 0.17$ | $0.57 \pm 0.12$ | $0.67 \pm 0.31$ | $0.72 \pm 0.53$ | $0.78 \pm 0.74$ | $0.82 \pm 0.96$ |
| Special | $0.58 \pm 0.16$ | $0.64 \pm 0.11$ | $0.70 \pm 0.31$ | $0.72 \pm 0.53$ | $0.79 \pm 0.74$ | $0.84 \pm 0.96$ |
| Phase | $0.60 \pm 0.15$ | $0.65 \pm 0.10$ | $0.72 \pm 0.30$ | $0.78 \pm 0.52$ | $0.81 \pm 0.73$ | $0.84 \pm 0.95$ |
| Complex | $0.63 \pm 0.20$ | $0.65 \pm 0.12$ | $0.75 \pm 0.36$ | $0.81 \pm 0.52$ | $0.79 \pm 0.74$ | $0.83 \pm 0.96$ |
| Multiplication (antithetical) | $0.56 \pm 0.15$ | $0.59 \pm 0.12$ | $0.67 \pm 0.32$ | $0.71 \pm 0.54$ | $0.79 \pm 0.75$ | $0.81 \pm 0.97$ |
| Special (antithetical) | $0.56 \pm 0.15$ | $0.65 \pm 0.12$ | $0.70 \pm 0.31$ | $0.73 \pm 0.52$ | $0.78 \pm 0.74$ | $0.81 \pm 0.96$ |
| Phase (antithetical) | $0.61 \pm 0.16$ | $0.66 \pm 0.11$ | $0.73 \pm 0.30$ | $0.77 \pm 0.52$ | $0.81 \pm 0.74$ | $0.86 \pm 0.95$ |
| Complex (antithetical) | $0.60 \pm 0.18$ | $0.66 \pm 0.14$ | $0.77 \pm 0.32$ | $0.76 \pm 0.52$ | $0.80 \pm 0.74$ | $0.84 \pm 0.96$ |

## A.2 EXPERIMENTAL DETAILS

For the simulated tasks, we generate training data of size 10,000 within the range $[-1, 1]$, and test data of size 2,000 within ranges $[-1, 1]$, $[0, 1]$, and $[-1, 0]$, respectively. The detailed KAN model configurations are provided in Table 7.

Table 7: Model specifications for simulated datasets. **Width** denotes the width of layers. **Grid** is the number of grid intervals. **Order** is the order of piecewise polynomials. **Lamb** indicates the overall penalty strength. **Optimizer** is the optimizer used for training.

| Task | Width | Grid | Order | Lamb | Optimizer |
|---|---|---|---|---|---|
| Multiplication | [2,5,1] | 3 | 3 | 0 | |
| Special | [2,5,1] | 5 | 3 | 0.1 | LBFGS |
| Phase | [3,5,1] | 5 | 3 | 0 | |
| Complex | [2,5,1] | 5 | 3 | 0 | |

For the real-world tasks, we evaluate three representative datasets from different domains: **Census-income** (tabular classification) (Kohavi, 1996), **MNIST**[3] (computer vision), and **Airbnb**[4] (mixed-type regression). ShapKAN, Vanilla KAN, and DropKAN share the same KAN architectures listed in Table 8.

For comparison, we also construct MLP baselines with the similar parameter scale of the corresponding KANs. The detailed configurations are shown in Table 9. In addition, we apply the unstructured $L_1$ pruning module provided in PyTorch[5].

---

[3] https://yann.lecun.org/exdb/mnist/index.html
[4] https://www.kaggle.com/datasets/arianazmoudeh/airbnbopendata
[5] https://pytorch.org/docs/stable/generated/torch.nn.utils.prune.l1_unstructured.html

Table 8: KAN model specifications for real-world datasets. **# Param** denotes the number of parameters. **Width** is the width of each layer. **Grid** is the number of grid intervals. **Order** is the order of piecewise polynomials. **Lamb** indicates the penalty strength. **Optimizer** is the training optimizer.

| Task | # Param | Width | Grid | Order | Lamb | Optimizer |
|---|---|---|---|---|---|---|
| Census-income | 8796 | `[40,12,6,4,2]` | | | | |
| MNIST | 17614 | `[49,16,12,10,10]` | 3 | 3 | 0 | LBFGS |
| Airbnb | 4714 | `[15,10,8,6,4,1]` | | | | |

Table 9: MLP model specifications for real-world datasets. **# Param** denotes the number of parameters. **Hidden Layer** gives the width of hidden layers. **Optimizer** is the training optimizer.

| Task | # Param | Hidden Layer | Optimizer |
|---|---|---|---|
| Census-income | 8,698 | `[72, 64, 16]` | |
| MNIST | 15,866 | `[128, 64, 16]` | Adam |
| Airbnb | 4,625 | `[90, 32, 8]` | |

LLM USAGE DECLARATION

We used Claude (Anthropic) as a writing-assistance tool to improve grammar and clarity during manuscript preparation. All research ideas, designs, and analyses were conducted by the authors, who take full responsibility for the accuracy and integrity of the content.

