# OpenReview forum: "Shift-Invariant Attribute Scoring for Kolmogorov-Arnold Networks via Shapley Value"
_ICLR.cc/2026/Conference — Submitted to ICLR 2026_

### Official Review · Reviewer_j4R9 · 2025-10-23

**Soundness:** 2
**Presentation:** 3
**Contribution:** 1
**Rating:** 2
**Confidence:** 3

**Summary:**

This paper proposes a principled and interpretable pruning framework for KANs, which are known for their spline-based activation functions and symbolic interpretability. Traditional magnitude-based pruning methods fail on KANs because node importance becomes unstable under input coordinate shifts. To address this, the authors formulate neuron attribution in KANs as a cooperative game and apply Shapley values to quantify each neuron’s true contribution in a shift-invariant manner. They further introduce efficient approximation techniques using permutation and antithetic sampling, along with a bottom-up multi-layer pruning algorithm. Extensive experiments on both synthetic and real-world datasets demonstrate that ShapKAN achieves more consistent neuron importance rankings, preserves predictive accuracy after pruning, and improves symbolic recovery of ground-truth functions compared to vanilla KANs and DropKAN. Overall, ShapKAN enhances the interpretability and robustness of KANs while enabling effective model compression for practical deployment.

**Strengths:**

1.	This paper presents a coherent framework that combines Shapley value for KAN architecture. The formalization of neuron contribution as a cooperative game is conceptually sound.
2.	The proposed ShapKAN method addresses a clear technical issue in KANs. The authors also propose an approximate algorithm to reduce the time overhead, demonstrating a very comprehensive consideration.
3.	This paper is well-written and easy to follow.

**Weaknesses:**

1.	The paper’s contribution is incremental. The use of Shapley values for assessing feature or neuron importance has been extensively studied in interpretability research . This work primarily adapts those ideas to the specific context of KANs without introducing a new theoretical or algorithmic insight.
2.	The method is tightly coupled with the structure of Kolmogorov–Arnold Networks, which are not widely adopted and remain primarily of experimental interest. As a result, the practical impact of improving pruning for KANs is limited. The authors do not discuss how the approach might generalize to other neural architectures or broader model types.
3.	While the framework is formulated using cooperative game theory, the paper mainly relies on established Shapley properties and known approximation results. There is no new theoretical guarantee or analysis demonstrating why Shapley-based pruning should be superior for function-based models like KANs, which rely on spline activations.
4.	The empirical validation is based on relatively simple datasets (e.g., MNIST, Census, Airbnb), where KANs offer no clear advantage over standard MLPs. The observed improvements in performance metrics (such as RMSE and accuracy) are modest and within expected variance ranges. Comparisons are restricted to KAN-specific baselines (Vanilla KAN, DropKAN), without benchmarking against more general or recent pruning and interpretability methods.
5.	The paper claims enhanced symbolic interpretability but evaluates it only on a few toy functions with known analytic forms (e.g., multiplication, Bessel-based functions). The improvements shown in symbolic recovery are minor and may not conclusively demonstrate superior interpretability.
6.	Although the paper introduces permutation and antithetic sampling to approximate Shapley values efficiently, experiments are conducted only on small-scale KANs. The feasibility of this approach for larger or deeper networks remains unclear, and no complexity or runtime analysis is provided for broader settings.

**Questions:**

Please refer to the Weaknesses section above.

---

### Official Review · Reviewer_Qxw1 · 2025-10-23

**Soundness:** 3
**Presentation:** 2
**Contribution:** 2
**Rating:** 2
**Confidence:** 3

**Summary:**

The paper proposes a method named ShapKAN for pruning KAN networks by computing the Shapley value to evaluate node importance. The computed Shapley values are then used to prune the network, starting from the bottom layers and proceeding up to the top layers. The authors evaluate the proposed approach on both synthetic and real-world datasets. The results show that ShapKAN can preserve important nodes and enhance both the performance and the interpretability of KAN. Additionally, the method is claimed to show stability under covariate shifts.

**Strengths:**

-The paper employs the Shapley value to solve the problem of finding the unimportant nodes of a Kolmogorov-Arnold network for pruning, which provides a unique solution with desired and theoretically supported properties.

-Apart from a few typos, the paper is clear and easy to follow.

-The results show improvement using ShapKAN over the competing approaches.

**Weaknesses:**

-The main weakness, which is not properly discussed in the paper, is the burden of computing the Shapley values in real-life scenarios. The employed datasets are trivial and small. In many cases, similar small-scale challenges can be solved efficiently using an interpretable machine learning algorithm. One of the main selling points of the paper is that it aids in maintaining interpretability in critical domains, e.g., clinical research. Therefore, I wonder how the proposed approach performs in a more challenging setting where there is a high number of variables that require a significantly bigger network architecture to maintain acceptable predictive performance. This can be directly addressed by analysing the computational complexity of the proposed approach. Currently, the authors provided recordings of the execution time in seconds, but that is for a small network architecture. It is also mentioned that the unbiased estimator converges at a standard rate of $O(1/\sqrt{m})$, however, this is computed for every node in the computational graph.

-The authors claim the proposed approach is consistent under covariate shifts. However, it is not extensively evaluated in the experiments, except in Section 4.1, where it is mentioned that:
>“Training data are sampled from N (0, 1) with random seeds, restricted to input ranges [−1, 1]. Test data are generated in three ranges: [−1, 1], [−1, 0], and [0, 1], to simulate covariate shift”

In my view, the shift-invariance is a main claim that has to be tested more extensively in the experiments.

**Questions:**

1- Is there a mistake in equation 5? $O_{l,i} = \underset{j}{max}(|\phi_{l−1,i,k}|_1)$, but where is $j$ in the equation?

2- Is it possible to clarify Figure 2? How to interpret the numbers?

3- In Figure 4, it looks like convergence with and without antithetic permutations has the same rate.

---

### Official Review · Reviewer_jjhq · 2025-10-28

**Soundness:** 2
**Presentation:** 2
**Contribution:** 2
**Rating:** 2
**Confidence:** 4

**Summary:**

This paper proposes ShapKAN, a pruning framework for Kolmogorov-Arnold Networks using Shapley values to score neuron importance. The key claim is shift-invariance: importance rankings remain stable when input domains change, unlike magnitude-based methods. The authors apply permutation sampling with antithetic pairs for approximation and test on four synthetic functions plus three real-world datasets. Results show improved symbolic recovery and test performance compared to vanilla KAN pruning and DropKAN.

**Strengths:**

1. The paper identifies a real problem. Fig 1 and Fig 2 demonstrate that vanilla KAN pruning yields inconsistent neuron rankings across input ranges. This instability matters for KAN's interpretability goals. The 50-run experimental design in sec 4.1 provides solid evidence.

2. Experiments are comprehensive. Four synthetic tasks test symbolic recovery under covariate shift (sec 4.3). Three real-world datasets span different domains (sec 5). The comparison includes DropKAN, vanilla KAN, and MLP baselines. Results consistently favor ShapKAN across metrics.

3. The approximation analysis is thorough. Sec 4.4 validates permutation sampling convergence. Fig 4 shows bias decreases as expected. Table 5 and 6 demonstrate computational efficiency remains reasonable even with large sampling sizes.

**Weaknesses:**

1.The coalition value function v(S) in eq 7 lacks precise definition. How do you compute KAN_S(x) when neurons are removed? The forward pass requires summing over all neurons in eq 2. Removing neurons changes the input distribution to downstream layers. do removed neurons output zero? Are edges reweighted?


2. Shift-invariance needs formal definition. You claim the method is shift-invariant but never define what transformations should preserve importance. The experiments test on different input ranges but you train on the full range. Why should we expect importance to be identical on [0,1] vs [-1,0] for the same function? Consider f(x) = ReLU(x). This function genuinely behaves differently on negative vs positive domains. Should importance change or not?

3. The theoretical justification for Shapley values in KANs is a bit weak. Sec 3.1 claims neurons form a cooperative game but KAN layers are additive (eq 2). Each edge φ_i contributes independently. There is no synergy term where v(S∪{i}) - v(S) depends on S. Shapley values are designed for games with interaction effects. In additive settings they reduce to individual contributions. Why is the exponential computational cost of coalition enumeration justified? Could normalized magnitude achieve the same shift-invariance more efficiently?


4. The symbolic recovery evaluation is superficial. Sec 4.3 provides one example per task. Tab 3 and 4 show recovered functions but no systematic evaluation. How often does ShapKAN recover correct symbolic forms across multiple runs? The paper claims this is crucial for interpretability but does not measure it rigorously.


5. The scope is unclear. You claim contributions to KAN interpretability and compression (sec 2). But sec 5 focuses entirely on prediction accuracy. Fig 5 shows performance vs pruning ratio but not symbolic recovery quality. Is the goal compression for deployment or interpretability for science?

**Questions:**

1. Eq 7 defines v(S) = E[KAN_S(x)]. Walk through a concrete example. Suppose layer l has 5 neurons and S={1,3,5}. How exactly do you compute the forward pass? What values do neurons 2 and 4 take? Please provide implementation pseudocode.

2. Sec 4.2 shows pruning decisions differ between methods. Did you verify these pruned models after retraining? Tab 2 reports test RMSE but are these from the same pruning decisions shown in Fig 2? The connection between stable rankings and final performance needs direct demonstration.

3. You avoid applying ShapKAN to the input layer (end of sec 3.3). But feature importance is a major application of Shapley values. Why exclude this?

---

### Official Review · Reviewer_tdwn · 2025-10-30

**Soundness:** 2
**Presentation:** 3
**Contribution:** 2
**Rating:** 2
**Confidence:** 4

**Summary:**

The paper proposes the usage of Shapley-Values to calculate importance scores for each node inside a KAN network. Therefore the acquired importance scores for each node are shift-invariant, which the authors claim is beneficial for pruning the network to only the most important nodes.

Furthermore the authors developed a bottom-up pruning algorithm, in which the contribution scores are calculated either by exact Shapley-Values for smaller layers or by Shapley-Approximations for wider layers. Nodes with low importance scores can then be removed in each layer for pruning the network. The selection of unimportant nodes can be done in different ways, to either achieve a user-specified total number of removed nodes per layer, remove only nodes which don't surpass a given ratio of importance in the current layer, or by simply setting a threshold for minimum node importance per layer.

The authors have shown in simulated function-fitting tasks that their proposed way of calculating importance-scores yields more consistent rankings of nodes across different shifted domains than traditional KAN importance scores.

**Strengths:**

While Shapley-Values are a well-established concept in computer science and machine learning, the usage of Shapley-Values for calculating node-importance scores for KANs is a novel and creative combination with very good motivation. The authors introduce and motivate the concept of Shapley-Values in great detail.

It is shown that the ordering of nodes based on their importance in the single hidden layer is more consistent for simulated function-fitting tasks and the results are visualized in an easy to understand overview in figure 2.

The anonymously published source-code supports open science and improves clarity of the paper.

Influences of bad luck on the experiments due to random seeds and randomly sampled data are eliminated by 50 repetitions of each experiment, leading to reliable results. Also the experimental setup is clearly stated including all the hyper-parameters in the appendix.

**Weaknesses:**

The biggest weakness of the paper is the limited number and scope of experiments. The simulated function-fitting experiments are all performed in a KAN with just one hidden layer, which contains only five nodes. While this is helpful for easier analysis of the proposed method, it is unclear how well it performs in deeper networks. Other experiments on real-world datasets, such as Census, MNIST and Airbnb, utilize deeper and wider KANs, however there is only a very limited number of experiments on real-world benchmarks and their results are inconclusive. In section 5 the figure 5 is described textually by "As illustrated in Figure 5, ShapKAN consistently outperforms Vanilla KAN in generalization.". This statement is not at all supported by figure 5 and just incorrect, as the red and green lines for ShapKAN and vanilla KAN respectively are overlapping, almost identical and show no clear winner. Other than figure 5 there are no further evaluations on real-world datasets and the section 5 on real-world benchmarks spans only two short paragraphs with very little information.

The next biggest weakness is the formulation of importance scores in vanilla KAN in section 2.2. While the given formulas and descriptions are correct for vanilla KAN 1.0 (Liu et al., 2024b) the same authors have revisited the calculation of importance scores in their KAN 2.0 paper (Liu et al., 2024a) by using variance / standard deviation rather than L1 scores. Although both papers are cited in this work, the improved way of calculating attribution scores in KAN 2.0 is not mentioned. Even worse, the anonymously published source code utilizes KAN 2.0 instead of KAN 1.0 and therefore the mentioned L1 scores for importance calculations are not even applied in experiments. Furthermore the original importance score calculation of vanilla KANs (1.0 and 2.0) can attribute nodes and edges, whereas the proposed Shapley-Values are only applied on a node-level.

Additionally the statement in the Introduction " [...] PyKAN (Liu et al.,2024a) overwrites its cached data during each forward pass, causing the same function to yield inconsistent importance scores across domains [...]" is true, however this is desired behavior and not a disadvantage. The vanilla PyKAN provides explanation / importance scores for the data fed into the network during the most recent forward pass. Therefore different subsets of the dataset can and should lead to highly different importance scores, as e.g. one feature might be very important in one subset and completely unimportant in another subset.

The relevance of shift-invariance for importance scores remains unclear and potential alternatives, such as simply normalizing the current batch or the whole dataset, are not discussed at all.

**Questions:**

Why is shift-invariance helpful / desired for the importance scores?

Why do we want similar importance scores for different / shifted inputs? In your example in figure 1 one would expect to have highly different importance scores for hidden neurons when x2>0 and x2<= 0, because in the latter case the feature x1 does not influence the model output at all and is therefore completely unimportant, since the model output is always 0 when x2 <= 0.

Why not simply use batch normalization or normalize the entire dataset so that covariate shift between training- and validation-split (on which pruning is performed) is less of a problem?

How does the formulation for importance scores using variance (KAN 2.0) instead of L1 scores or magnitude (KAN 1.0) change your comparison? Variance should already introduce shift-invariance and therefore make your contribution less significant?

---

### Meta-Review · Area_Chair_iwmd · 2026-01-05

**Summary:**

The paper proposes a pruning approach for Kolmogorov-Arnold Networks (KANs) based on Shapley values. All four reviewers gave this submission a score of 2 (reject). Some of the most important issues that were raised are:
- Limited experimental scope with focus on shallow networks.
- Unclear when and why is it desirable that importance scores are invariant to input shifts (which also lacks a formal definition as pointed out by Reviewer jjhq), since some features may genuinely have different across different ranges.
- Scalability concerns

Reviewer tdwn also raised implementation inconsistencies between KAN 1.0 and KAN 2.0 which further cast doubt on some of the results. Reviewer j4R9 was critical of the paper's novelty and also commented that KANs themselves have limited adoption.

The authors did not respond to any of the questions or issues raised. I agree with the unanimous recommendation for rejection.

**Reviewer Concerns:**

No rebuttal.

**Reviewer Scores:**

No rebuttal.

---

### Decision · Program_Chairs · 2026-01-26

Reject